# Impacts of COVID-19 on sexual behaviors, HIV prevention and care among men who have sex with men: A comparison of New York City and Metropolitan Atlanta

Steven M. Goodreau[1,2]*, Kevin P. Delaney[3], Weiming Zhu[3], Dawn K. Smith[3], Laura M. Mann[4], Travis H. Sanchez[4], Deven T. Hamilton[2], Karen W. Hoover[3]

1 Department of Anthropology, University of Washington, Seattle, Washington, United States of America,
2 Center for Studies in Demography and Ecology, University of Washington, Seattle, Washington, United States of America, 3 Division of HIV Prevention, National Center for HIV, Viral Hepatitis, STD and TB Prevention, Centers for Disease Control and Prevention, Atlanta, Georgia, United States of America,
4 Department of Epidemiology, Rollins School of Public Health, Emory University, Atlanta, Georgia, United States of America

* goodreau@uw.edu

**Data Availability Statement:** This study uses data provided by four outside sources to conduct

## Abstract

The COVID-19 pandemic has disrupted HIV prevention, care, and transmission opportunities. This likely varies by geography, given differences in COVID-19 burden and mandates over time, and by age, given different likelihoods of severe COVID-19 consequences. We consider changes in sexual behavior, HIV testing, pre-exposure prophylaxis (PrEP) use and antiretroviral therapy (ART) use among men who have sex with men (MSM) over the first year of the COVID-19 epidemic, comparing the Atlanta metropolitan area and New York City (NYC). We use two continuous time-series datasets and one panel dataset, assessing changes over time within city and comparing across cities, and disaggregate major findings by age. For clinical results, ART use showed by far the smallest reductions, and testing the largest. Disruptions occurred concurrently between cities, despite the major wave of COVID-19, and government mandates, occurring later in Atlanta. Test positivity increased in NYC only. In both cities, younger MSM saw the greatest reductions in testing and PrEP use, but the smallest in sexual behavior. Reduced clinical service usage would be unconcerning if stemming solely from reductions in exposure; however, the patterns for young MSM suggest that the COVID-19 epidemic likely generated new conditions for increased HIV transmission, especially in this cohort.

## Introduction

Although gay, bisexual and other men who have sex with men (MSM) continue to experience the majority of HIV burden in the United States, incidence within this community is declining, with new diagnoses down nearly 10% from 2015 to 2019 [1]. This is largely due to increases in both pre-exposure prophylaxis (PrEP) and viral suppression from antiretroviral

secondary analyses. Our data use agreements preclude us from sharing these data in order to maximize the sources' ability to ensure patient/respondent confidentiality and safety. Each source has its own process for additional qualified users to obtain data after a process of scientific review to balance safety concerns with research needs. Links to these processes are: IQVIA (https://www.iqvia.com/solutions/real-world-evidence/real-world-data-and-insights), Labcorp (https://www.labcorp.com/organizations/data), Quest Diagnostics (https://www.questpharmasolutions.com/pharma/healthcare-analytics-solutions/solutions/pharma-analytics/quest-data-insights-platform/), and the American Men's Internet Survey (AMIS, https://emoryamis.org/data-requests/). The authors did not have any special privileges in accessing the data. We provide all of our analysis code on GitHub (https://github.com/UW-CAMP/CovidHIV_NYCvATL) so that interested readers may understand the full detail of our analytical methods.

**Funding:** This work was supported by the U.S. Centers for Disease Control and Prevention's National Center for HIV/AIDS, Viral Hepatitis, STD, and TB Prevention [cooperative agreement U38-PS004650, PI Erika Martin]. Additional support was provided by a research grant from NIAID [R01 AI138783, PI Samuel Jenness] and a research infrastructure grant from NICHD to the UW Center for Studies in Demography and Ecology [grant number P2C HD042828, PI Sara Curran]. As part of the cooperative agreement model, research scientists affiliated with the CDC contributed to the design of the study through a series of collaborative meetings. They also approved the decision to submit the manuscript for publication, and CDC staff scientists reviewed and approved the final manuscript. Those individuals who played significant roles in the scientific development of the project are listed as co-authors. CDC played no additional role beyond these contributions. The findings and conclusions in this report are those of the authors and do not necessarily represent the official position of the Centers for Disease Control and Prevention. No part of NIH, including the funders NIAID and NICHD, played a role in study design, data collection and analysis, decision to publish, or preparation of the manuscript.

**Competing interests:** The authors have declared that no competing interests exist.

therapy [ART, 2,3]. Continued progress is critical to achieving Ending the HIV Epidemic in the US (EHE) initiative goals of reducing HIV incidence by 75% by 2025 and 90% by 2030 [4].

In early 2020, however, use of health services was disrupted, along with all other aspects of daily life, by the COVID-19 pandemic. Multiple surveys early in the pandemic found that US MSM reported reduced access to and usage of various HIV prevention and care services, including testing, PrEP and ART, in both national and local samples [5–10]. Variations in timing, methods, wording and populations make it difficult to directly compare all studies, however. Observed reductions in HIV testing and PrEP use are presumably driven by some combination of reductions in demand (given behavioral changes) and access [given both stay-at-home orders and the repurposing of HIV/STI staff and services to COVID-19 prevention, 11]. ART should not see similar demand reductions among those already diagnosed, although new prescriptions might drop along with testing.

Multiple studies early in the pandemic also confirmed that social distancing generated some period of reduced sexual contacts overall, likely reducing HIV transmission opportunities. Estimates cluster around ⅓-½ of men reducing contacts [8,12,13], although one study found that 79% of respondents had done so [6], another found an increase in partnering [14], and one found no change in casual partner numbers but a reduction in condomless anal intercourse [CAI, 15] Some changes varied by whether a partner was main and/or co-resident [6,12,13], an important distinction since estimates suggest that a sizeable fraction of HIV transmissions among MSM occur between main partners [16,17]. The simultaneous reduction in sexual networking and in testing makes it difficult to know how much reductions in HIV diagnoses reflect parallel reductions in incidence, either nationally or in specific jurisdictions.

The COVID-19 epidemic has, of course, not disappeared since those efforts to document initial disruptions. A series of papers explores the magnitude of changes in three HIV services (PrEP, ART and testing) over the first pandemic year, finding PrEP use to be most impacted [22% reduction relative to expected averaged over the year, 18], testing next [15%, 19], and ART least [2.5%, 20]. For PrEP and ART, sex, age and insurance type were significant predictors, and differences were seen by region, with greatest drops in the Northeast.

Indeed, the COVID-19 epidemic has been highly heterogeneous across the country in overall magnitude and timing of waves [21]. Jurisdictions have also varied widely in timing and intensity of social restrictions [22]. However, it is unclear the extent to which the timing and magnitude of specifically HIV-relevant behavioral and clinical changes align with local COVID burden and policies vs. national ones. Either could reasonably be hypothesized—for instance, studies of spatial mobility have found that all states examined underwent roughly simultaneous reductions in spatial mobility at the epidemic's start [23]; however, subsequent government policies generated sizeable (>20%) additional reductions [24]. That said, use of HIV services is not synonymous with spatial mobility reduction, especially during a time of expanded telemedicine and mail-order pharmacy use. Developing a sense of the patterns of change in different areas, including their overall similarities and differences, may help jurisdictions to better interpret their local HIV surveillance data.

COVID-19 also does not impact all ages equally, with severe morbidity and mortality highly concentrated among older persons. If reductions in health care service use were driven primarily by individual concerns over COVID-19 exposure risk, one might predict that older men would show the highest reductions. If, however, they were mostly due to facility closure or staff unavailability, one might expect similar effects across age. And younger MSM could actually show the greatest reductions experienced if they experienced greater challenges in navigating health care system changes. Given the US's decentralized health care systems, these patterns might vary across jurisdictions, with relevance for health departments trying to

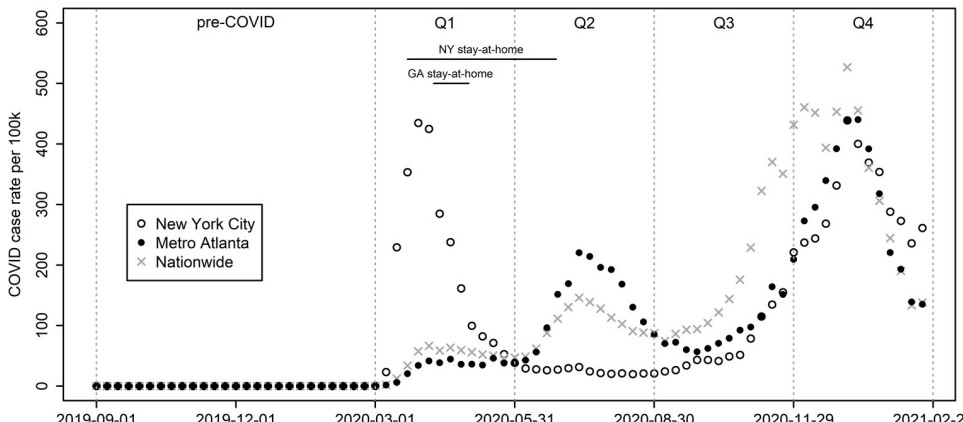

**Fig 1. Weekly COVID-19 diagnoses.** Sources: NYC Coronavirus Disease 2019 (COVID-19) Data Repository (https://github.com/nychealth/coronavirus-data). Georgia Department of Public Health Daily Status Reports (https://dph.georgia.gov/covid-19-daily-status-report). The Centers for Disease Control and Prevention COVID Data Tracker (for national numbers https://covid.cdc.gov/covid-data-tracker/#trends_dailytrendscases). US Census Bureau Vintage 2020 Population Estimates for the United States and States (https://www.census.gov/programs-surveys/popest/technical-documentation/research/evaluation-estimates.html). COVID-19 US state policy database [www.tinyurl.com/statepolicies, 25].

understand where the greatest unmet need for services has accumulated since the COVID-19 pandemic began.

This paper considers changes in sexual behavior, HIV testing, PrEP use and ART use over the course of the first year of the COVID-19 epidemic in two illustrative locations: the Atlanta metropolitan area (hereafter "Atlanta"), and New York City (NYC). These locations have distinct HIV prevention and care cascades; for example, the latest NHBS MSM round found sizeable differences in HIV prevalence (Atlanta = 33.3%, NYC = 18.4%), recent PrEP use (Atlanta = 21.6% of HIV-negative men; NYC = 32.1%) and current antiviral use (Atlanta = 87.2% of HIV-positive respondents; NYC = 95.2%). With COVID-19, the two locations also experienced large distinctions in timing of the first major wave and in scale and timing of resulting mandates (Fig 1). New York City (NYC) was the epicenter of the first US wave (Mar.-May, 2020), and imposed extensive restrictions, including the "New York State on Pause" stay-at-home order from Mar. 22-Jun. 27, 2020 [25]. Atlanta, in contrast, had a limited early outbreak, and Georgia's stay-at-home order began later (Apr. 3), and ended earlier (May 1), although a shelter-in-place order for medically fragile persons, including those with "poorly-controlled HIV", continued until Jun. 12 [25]. Atlanta then experienced a major summer wave along with much of the country, while NYC did not [26,27]. Finally, a third wave expanded throughout Autumn 2020 nationwide, including both locations, and peaking in early winter.

For this comparison, we examine three data sources: two continuous time-series datasets and one panel dataset with waves before and after COVID-19's arrival. We use both descriptive and inferential approaches to assess patterns of change over time within city and how these compare across cities. Finally, we disaggregate major findings by age group, and compare these age patterns by region.

## Methods

Our data sources have nationwide coverage; we use ZIP code of residence to limit to respondents in the five boroughs of NYC and 29 counties of the Atlanta-Sandy Springs-Alpharetta

Metropolitan Statistical Area. We chose NYC proper rather than MSA to keep the analysis within one state. For each datatype, we first consider overall metrics; for those showing meaningful differences over time and/or between cities, we further disaggregate by age.

## IQVIA data (PrEP and ART)

Our main source for PrEP and ART is the IQVIA Real World Data—Longitudinal Prescriptions Database (IQVIA). Methods for extracting prescription counts and defining continuous usage spells vs. new prescriptions have been described previously [18,20]. We consider three measures for PrEP and ART each, by week: number of ongoing prescription spells, number of spells initiated, and number terminated. We consider Sep. 1, 2019 through Feb. 29, 2020 as pre-COVID-19 data (26 weeks), and Mar. 1, 2020 through Feb. 27, 2021 as first-year COVID-19 data (52 weeks); we divide the latter into four 13-week quarters (Q1-Q4). We consider rates relative to the mean of the four weeks immediately prior to COVID-19 (Feb. 2 –Feb. 29, 2020); we chose these over absolute numbers since they are easier to compare across cities, and because the sources do not cover 100% of all relevant services in the US.

We first provide descriptive results, overall for our three metrics, and also by age for prescription prevalence. We then conduct interrupted linear time-series analyses on prescription prevalence based on the quarters described above. That is, we fit standard linear models with an intercept and single term for week (with 2019-09-01 as week 1), along with dummy terms for each of the four quarters after COVID onset, and quarters interacted with week. These latter two sets of terms provide measures of the intercept and slope for each quarter relative to the pre-COVID period.

Unfortunately, additional demographic information such as race/ethnicity had very large amounts of (likely non-random) missingness. Mode of transmission was also not distinguished, so we could not limit the analysis to MSM. We thus focus on trends for male patients as a whole, knowing that MSM represent the vast majority of males with incident HIV diagnoses (86%), prevalent infections (81%) and virally suppressed infections [84%, 1,28]. Moreover, PrEP use among heterosexual males is considered "negligible" [29]. This inclusion of all males further justifies our focus on relative rather than absolute numbers.

## Laboratory data (HIV testing)

Data were consolidated from reported daily testing results generated by two large national commercial laboratories: Labcorp and Quest Diagnostics. See supplement for technical details of data extraction. Results were based on specimen collection date, combined across laboratories, and again restricted to those with male gender living in NYC or Atlanta. We report weekly number of valid tests (again relative to Feb. 2–29, 2020) and positivity (proportion of valid tests with positive results). For interrupted time series, we added a quadratic (week-squared) term for HIV tests during Q1, given the strong pattern of decline and recovery within this single quarter. We also disaggregated test counts by age.

## AMIS (sexual behavior and additional clinical measures)

The American Men's Internet Survey (AMIS) is an online survey of US cisgender MSM 15 + years, conducted annually in fall/winter, with ~10,000 respondents/year [30,31]. Since many questions ask about behaviors in the last 12 months, we compare the 2019 and 2020 cycles within and across cities to identify changes; we refer to these as NYC-19, NYC-20, ATL-19, and ATL-20. Data collection occurred Sep. 19-Dec. 17, 2019 and Oct. 10, 2020-Jan. 5, 2021, respectively. We compare NYC-19 and ATL-19 to identify regional differences before COVID-19, then compare each area's 2019 survey to 2020 to identify levels of change during

COVID-19. We rely primarily on non-parametric tests (chi-squared, Fisher's exact, Mann-Whitney U-test) since our data are mostly nominal or ordinal and/or subject to rounding and outliers (e.g. number of partners). For metrics showing meaningful differences in change between cities, we further consider differences by age. Additional questions in 2020 asked how COVID-19 had impacted respondents' sexual behavior and HIV services use [8]; we compare these between cities, and use multiple logistic regression to identify predictors for reporting that the pandemic had caused a reduction in one's number of sexual partners.

### Ethics statement

Data on PrEP and ART were derived from a national large-scale prescription database, and data on testing from two national large-scale laboratory databases. All three were anonymized and reduced to a minimum set of non-identifiable data (e.g. sex, age, race) with minimum geographic granularity that precluded deductive disclosure. They were then provided to the CDC for research purposes. AMIS is a nationwide online research study whose participants initially underwent informed consent, which includes a statement that other researchers may receive grouped de-identified versions of the data for valid research purposes. The AMIS team require a research proposal and data use agreement before sharing. All data were fully anonymized before shared. The University of Washington Human Subjects Division reviewed the protocol for this secondary data analysis project, and confirmed that it did not qualify as human subjects research as defined by federal and state regulations given the anonymity of the data.

## Results

### IQVIA data (PrEP and ART)

Fig 2 shows three PrEP metrics over time: prevalence of PrEP prescriptions for men and numbers starting and ending PrEP spells, all relative to pre-COVID numbers. S1 Fig in S1 File and S1 Table in S1 File contain time-series metrics quantifying these effects on the absolute scale for prescription prevalence; the model fits NYC extremely well ($R^2$ = 0.990), and Atlanta moderately so ($R^2$ = 0.831). NYC saw a much larger relative decline in PrEP coverage over the year than Atlanta, with the former averaging 84.0% of pre-COVID PrEP prescriptions throughout Q2-4 (range 81.5–86.2%), and the latter averaging 95.8% (range 94.0–97.8%). While both underwent a comparably-sized decline in proportion of men *initiating* PrEP during Q1, NYC uniquely also underwent a 2-3-month period with more men *terminating* PrEP, corresponding with the large COVID-19 wave locally, and concomitant lockdowns. Thus, by far the largest change in slope for current PrEP prescriptions for any city or time period (S1 Table in S1 File) is NYC in Q1. Atlanta also experienced a Q1 decline in PrEP prescriptions, although much smaller in scale than NYC; by Q2, when Atlanta's COVID caseload expanded into a major wave, there was no comparable spike in PrEP termination nor dip in initiation. NYC returned to pre-COVID levels in initiation and termination, albeit at lower overall prevalence, by Q3, while Atlanta saw initiations and terminations both eventually increase above pre-COVID levels; collectively these trends caused prevalent prescriptions to surpass pre-COVID levels in Atlanta only. Fig 3 disaggregates PrEP prevalence by six age groups, revealing a consistent story by city: the youngest age groups saw the greatest declines in prescriptions. NYC 13-24-year-olds show the largest drop by far, leveling out at 60% of pre-COVID levels. Meanwhile, the oldest men saw the least decline, and in some cases slight increases.

ART coverage over time was much flatter than for PrEP, and also more similar between cities (Fig 4 and S2 Fig in S1 File, S2 Table in S1 File). Here the quarterly time series fits the data well in both Atlanta (0.973) and NYC (0.947); NYC averaged 98.9% of pre-COVID prescriptions during Q2-4 (range 98.3–99.7%) and Atlanta 101.3% (99.9–103.1%). Although nearly all

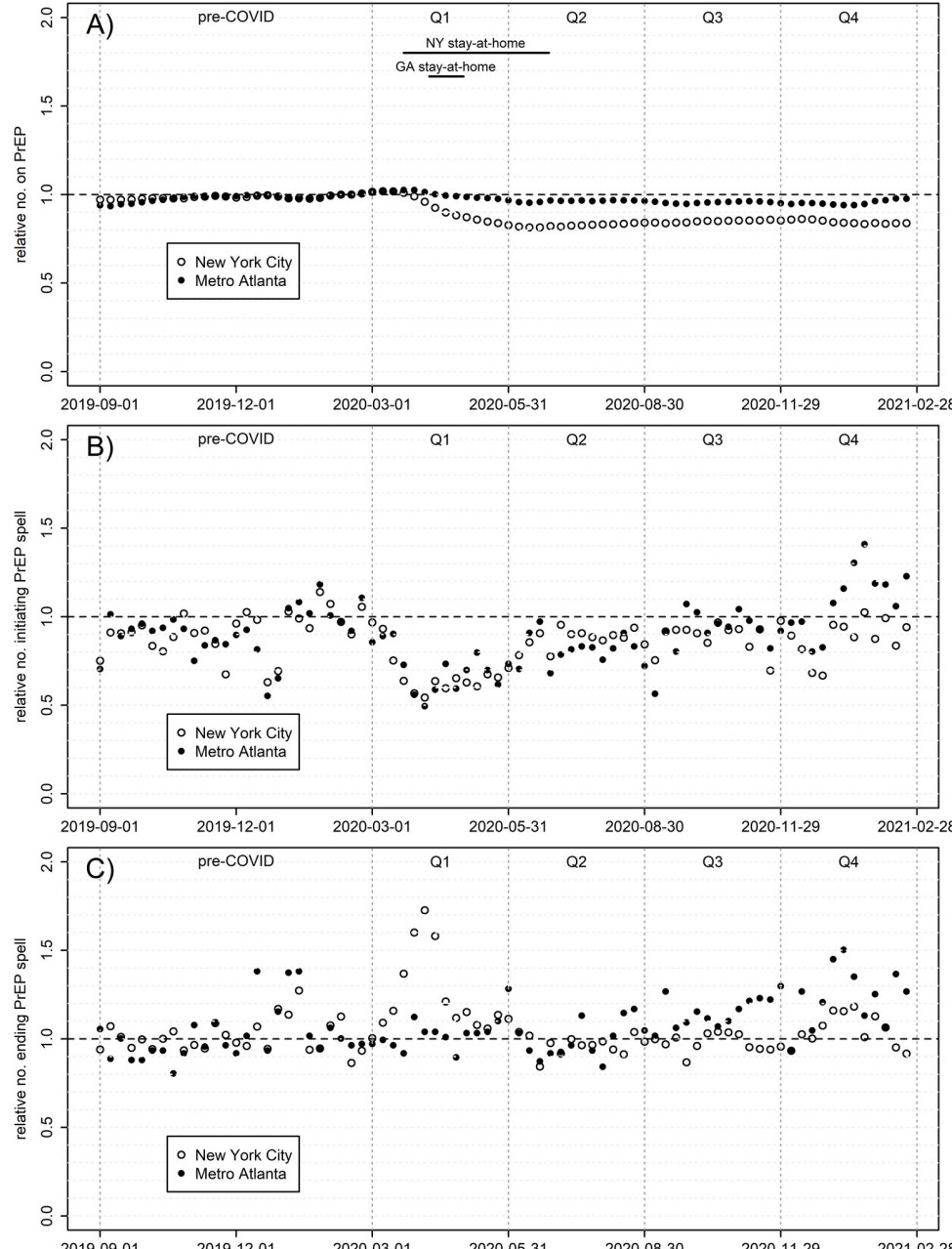

**Fig 2. PrEP prescriptions among males, by week, relative to the weekly mean Feb. 2 –Feb. 29, 2020.** A) Relative prevalence of ongoing prescription spells; (B) relative number initiating a new prescription spell; C) relative number terminating a prescription spell.

quarter-specific slopes are significant (partly due to larger sample sizes), effect sizes are all far smaller than for PrEP. Both cities saw virtually no effect of COVID-19 on ART terminations, and small similarly-sized declines in number initiating ART, from mid-Q1 through early Q4. Disaggregating by age (S3 Fig in S1 File) shows a similar ordering as for PrEP, where prescriptions relative to pre-COVID were lowest for younger cohorts and highest for older.

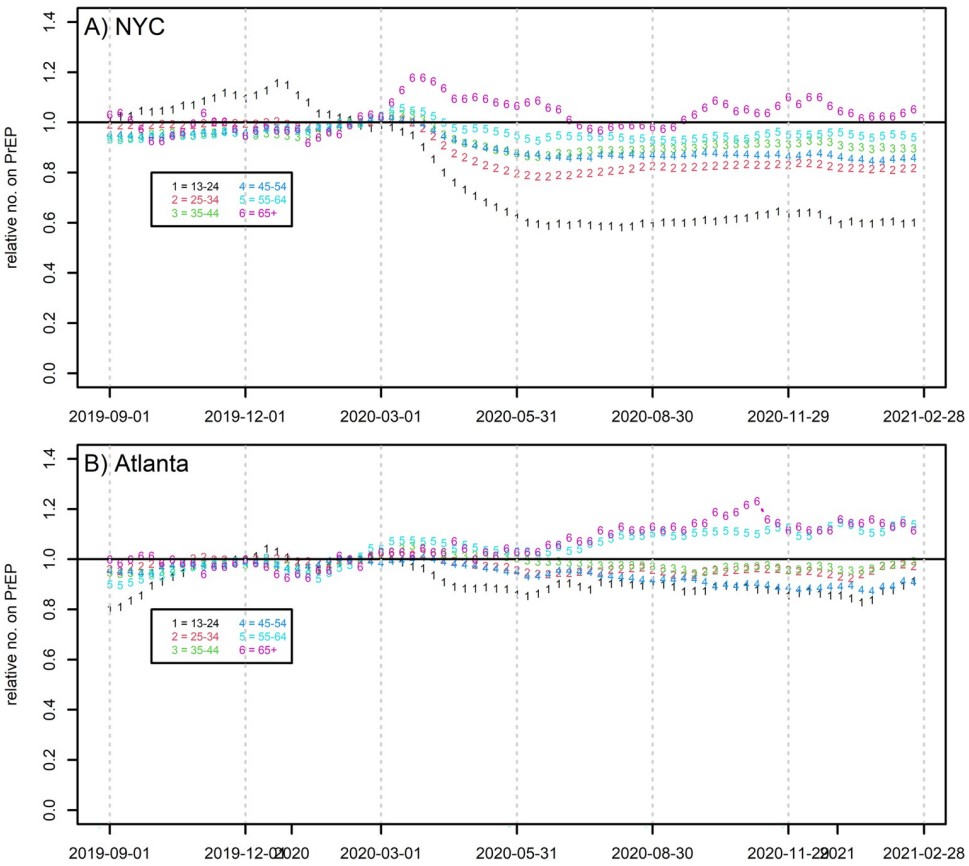

**Fig 3. Prevalence of active PrEP prescriptions by age, relative to the weekly mean Feb. 2 –Feb. 29, 2020.** A) New York City B) Metro Atlanta.

## Laboratory data: Testing

Relative numbers of tests and absolute positivity rates are shown in Fig 5, and time-series metrics for the former in S4 Fig in S1 File and S3 Table in S1 File. Testing rate fluctuations are visible around major holidays. Both cities saw dramatic reductions in testing that are similarly timed to COVID-19's emergence domestically, but of different magnitude; NYC bottomed out at only 13% of pre-COVID levels during Apr. 5–11, 2020, and Atlanta at 33% the same week. One might expect a concurrent increase in positivity given higher selectivity towards men with greatest HIV exposure; however, only NYC saw this, with positivity rates rising 2-3-fold to levels similar to Atlanta at the time. Both plateaued at a new level in early Q2, remaining consistently between 80–90% of pre-COVID counts for all subsequent weeks, again with the exception of major holidays. Atlanta saw a small but significant linear decline in positivity equivalent to falling 35% of one percentage point over the first year of COVID (analysis not shown); NYC exhibits no comparable trend. Changes do not show an age trend (S5 Fig in S1 File) as they did for PrEP and ART. For example, during the period of minimum testing (Mar. 29-May 5, 2020), all age groups averaged 17–21% of pre-COVID levels in NYC, with no clear ordering by age; Atlanta's range was 34–39%, with one outlier (15–25 year-olds, at 52%).

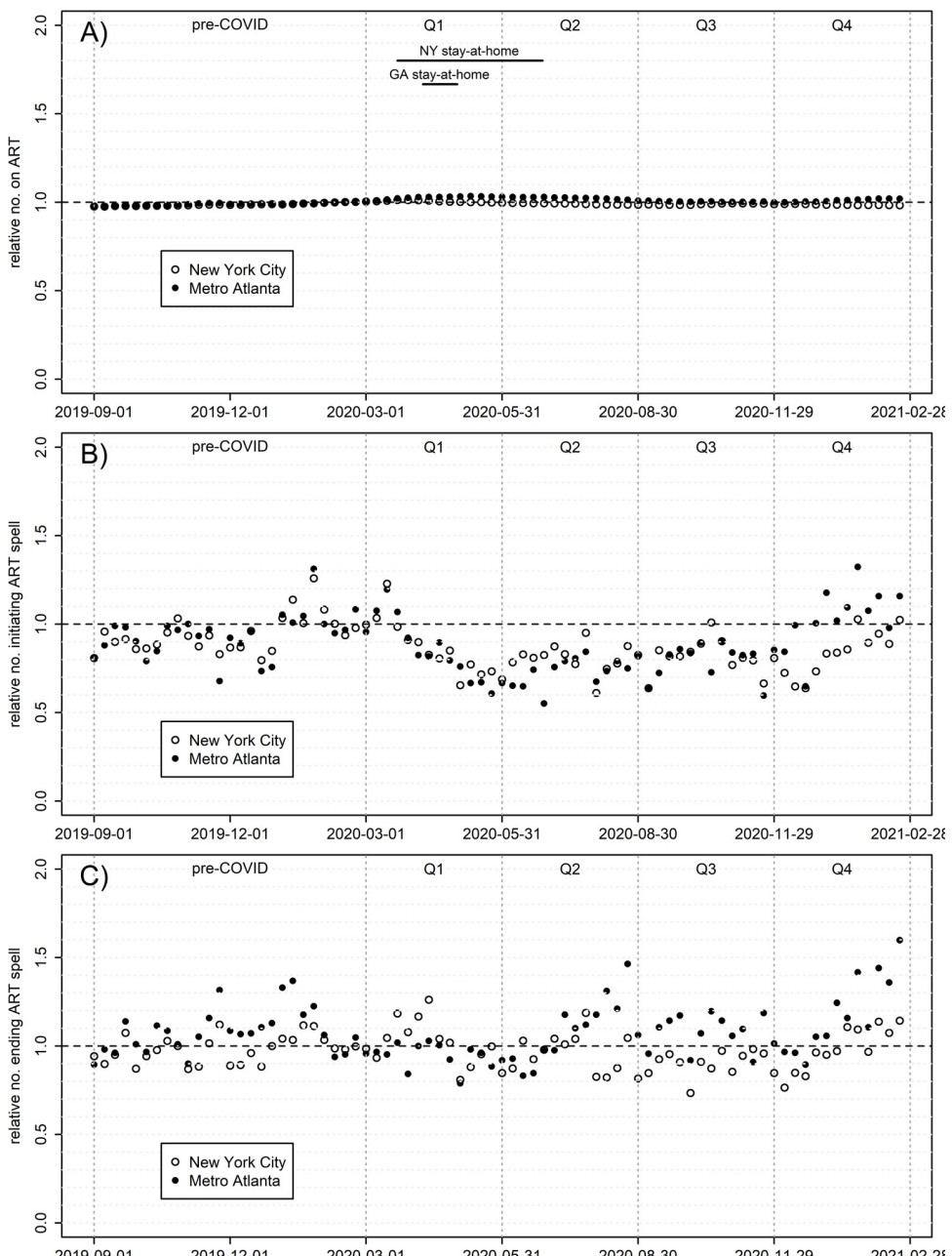

**Fig 4. ART prescriptions among males, by week, rel. to weekly mean Feb. 2-Feb. 29, 2020.** (A) Relative prevalence of ongoing prescription spells; (B) relative number initiating a new prescription spell; (C) relative number terminating a prescription spell.

## AMIS

Table 1 details sample demographics and sexual behavior metrics across two waves of AMIS. Composition by race/ethnicity and HIV status differ significantly across city in 2019, unsurprisingly. No demographic measures varied significantly across year in NYC; in Atlanta the 2020 sample had higher variance in age and lower HIV positivity.

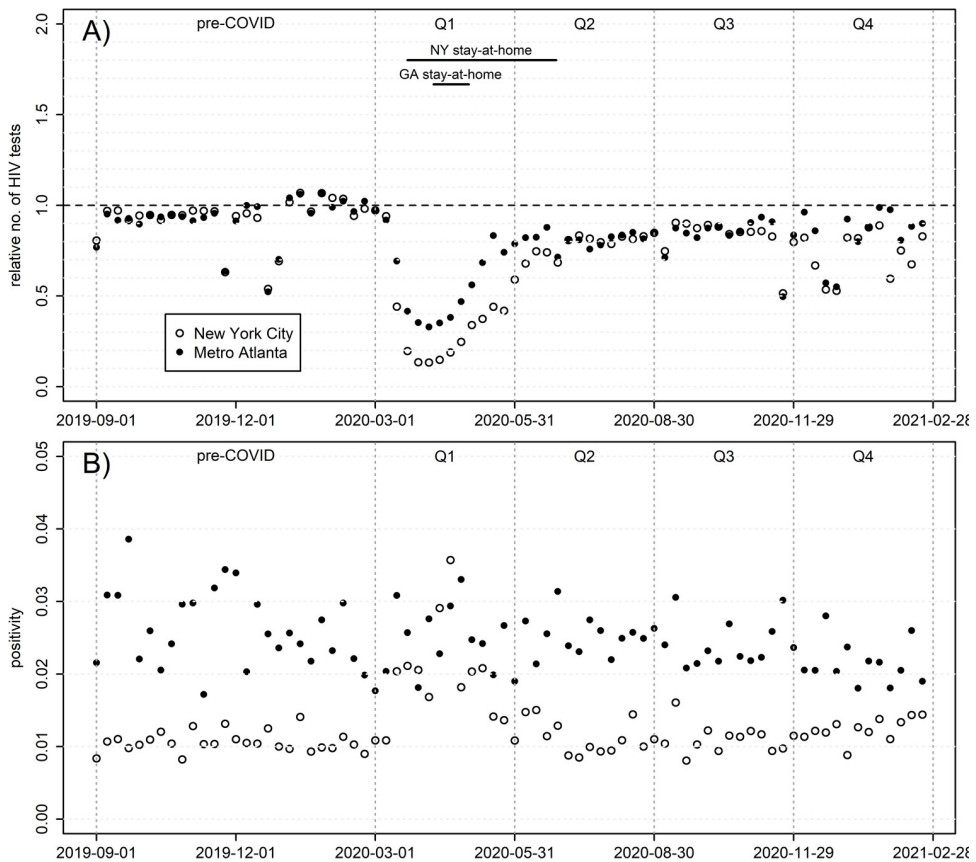

**Fig 5. HIV testing among males, by week.** A) Number of tests, relative to the weekly mean Feb. 2 –Feb. 29, 2020; B) positivity (proportion of valid tests with positive results).

**Anal intercourse (AI).**   Comparing the full distribution of number of AI partners in the last 12 months for ATL-19 and NYC-19 shows no significant difference (Mann-Whitney U-test; P = 0.21). Table 1 shows the distribution binned; in both cities, the proportion with 1 partner grew and those with >1 declined. In NYC this reduction occurred more from the middle of the distribution, and in Atlanta from the upper tail. Despite similarity in magnitude, changes in the distribution over time were statistically significant in Atlanta (P = 0.02), but not NYC (P = 0.10). Relatedly, the 90% quantile reduced in Atlanta (15 partners to 10) but not NYC (15 to 15). Among those reporting one partner in the year, the proportion who considered them their main partner increased in both cities, again significantly in Atlanta and not NYC, with larger effect size in the former. Disaggregating by age (S6 Fig in S1 File) helps clarify where the differences between cities lie. In Atlanta, both the 26–54 and 55+ age groups see significant shift downward in their distribution of AI partners (P = 0.04 and 0.05, respectively), while the 15-24-year-olds do not (P = 0.43); however, in NYC only the 55+ age group does (P = 0.02, vs. P = 0.34 for 18-24-year-olds and P = 0.83 for 25-54-year-olds).

**Condomless anal intercourse (CAI).**   Similar to all AI partners, comparing this distribution across cities for 2019 found no significant difference (Mann-Whitney U-test; P = 0.30), and cross-year differences were significant for Atlanta, but not NYC. In this case, however, disaggregating by age (Fig 6) shows the same pattern by city: significant reductions for men 55+ (P = 0.03 and 0.05 for Atlanta and NYC, respectively), but not other age groups. Across all four surveys, between ¼-⅓ of respondents reported having had some CAI with a partner who was

**Table 1. Measures from the American Men's Internet Survey of US men who have sex with men, 2019 and 2020.**

| | | New York City (NYC) | | Metro Atlanta (ATL) | | Significance tests | | | |
| --- | --- | --- | --- | --- | --- | --- | --- | --- | --- |
| | | 2019 | 2020 | 2019 | 2020 | NYC-19 vs. ATL-19 | NYC-19 vs. NYC-20 | ATL-19 vs. ATL-20 | NYC-20 vs. ATL-20 |
| | N | 292 | 385 | 433 | 360 | | | | |
| Age | 15–24 | 90 (30.8%) | 122 (31.7%) | 126 (29.1%) | 113 (31.4%) | P = 0.67 | P = 0.59 | **P = 0.04** | |
| | 25–54 | 171 (58.6%) | 218 (56.6%) | ***267 (61.7%)*** | ***196 (54.4%)*** | | | | |
| | 55+ | 31 (10.6%) | 45 (11.7%) | 40 (9.2%) | 51 (14.2%) | | | | |
| Race/ethnicity | NH Black | ***78 (27.5%)*** | ***78 (21%)*** | ***203 (47.9%)*** | ***147 (41.6%)*** | **P<0.01** | P = 0.29 | P = 0.15 | |
| | Hisp/Latinx | 63 (22.2%) | 92 (24.8%) | 46 (10.8%) | 32 (9.1%) | | | | |
| | NH White | 123 (43.3%) | 171 (46.1%) | ***148 (34.9%)*** | ***151 (42.8%)*** | | | | |
| | Other | 20 (7.0%) | 30 (8.1%) | 27 (6.4%) | 23 (6.5%) | | | | |
| Diag. with HIV | | 43 (14.7%) | 44 (11.4%) | ***101 (23.3%)*** | ***62 (17.2%)*** | **P = 0.01** | P = 0.25 | **P = 0.04** | |
| Num. AI partners | 0 | 24 (8.8%) | 40 (10.5%) | 51 (12.5%) | 45 (12.7%) | P = 0.21[a] | P = 0.10 [a] | **P = 0.02** [a] | |
| | 1 | ***57 (20.8%)*** | ***105 (27.6%)*** | 86 (21.1%) | 104 (29.3%) | | | | |
| | 2–5 | ***113 (41.2%)*** | ***131 (34.5%)*** | 166 (40.8%) | 137 (38.6%) | | | | |
| | 6+ | 80 (29.2%) | 104 (27.4%) | ***104 (25.6%)*** | ***69 (19.4%)*** | | | | |
| | 90th quantile | 15 | 15 | 15 | 10 | | | | |
| Prop. of those with one AI partner for whom it is a main partner | | ***71.8%*** | ***79.4%*** | ***64.9%*** | ***82.5%*** | P = 0.59 | P = 0.53 | **P = 0.02** | |
| Num. CAI partners | 0 | 84 (30.8%) | 103 (27.6%) | ***103 (25.4%)*** | ***110 (31.2%)*** | P = 0.30 [a] | P = 0.96 [a] | **P = 0.01** [a] | |
| | 1 | ***66 (24.2%)*** | ***114 (30.6%)*** | 110 (27.1%) | 104 (29.5%) | | | | |
| | 2–5 | 77 (28.2%) | 91 (24.4%) | 128 (31.5%) | 100 (28.3%) | | | | |
| | 6+ | 46 (16.8%) | 65 (17.4%) | ***65 (16.0%)*** | ***39 (11.0%)*** | | | | |
| | 90th quantile | 10 | 11.6 | 9 | 6 | | | | |
| Any discordant CAI | | 29.8% | 25.2% | 32.8% | 30.2% | P = 0.44 | P = 0.21 | P = 0.50 | |
| Impact of COVID on # of sexual partners | Decreased | | 144 (64.6%) | | 83 (41.7%) | | | | **P = 0.03** [b] |
| | Same | | 70 (31.4%) | | 102 (51.3%) | | | | |
| | Increased | | 9 (4.0%) | | 14 (7.0%) | | | | |
| Impact of COVID on use of condoms | Decreased | | 18 (8.1%) | | 13 (6.6%) | | | | P = 0.59 [b] |
| | Same | | 197 (89.1%) | | 178 (90.4%) | | | | |
| | Increased | | 6 (2.7%) | | 6 (3.0%) | | | | |
| Impact of COVID on getting HIV tested | Decreased | | 40 (22.2%) | | 18 (12.1%) | | | | P = 0.91 [b] |
| | Same | | 137 (76.1%) | | 127 (85.2%) | | | | |
| | Increased | | 3 (1.7%) | | 4 (2.7%) | | | | |

[a] Applied to whole distribution, not to bin values, using Mann-Whitney U-tests.

[b] Uses Fisher's exact test given the low numbers in at least one cell.

Comparisons use chi-squared tests unless otherwise noted.

***Bold italics*** = difference of at least five percentage points across years within city.

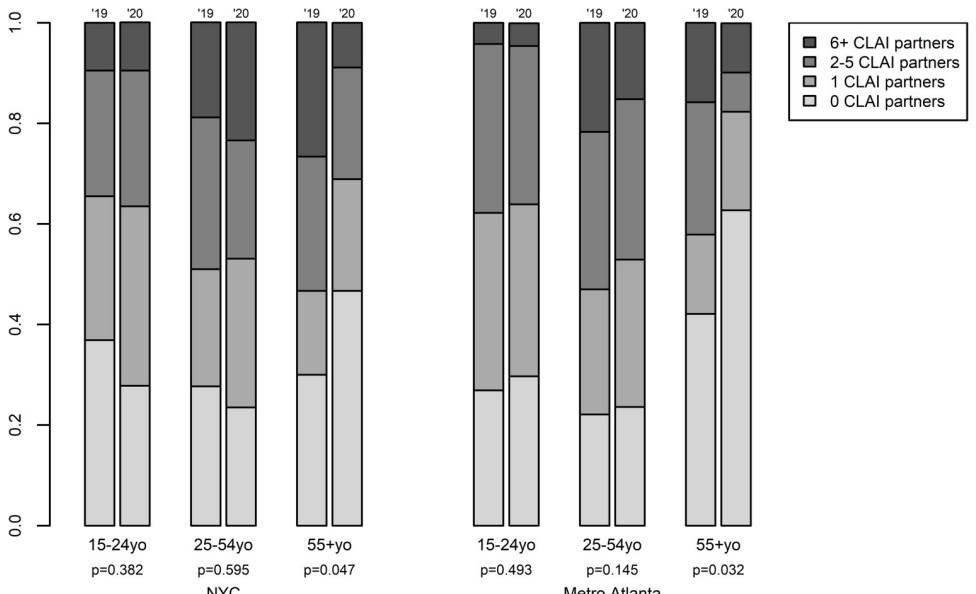

**Fig 6. Number of condomless anal intercourse (CLAI) partners in the last 12 months, by year, city and age.** Left-hand columns per pair = data from the 2019 round of AMIS; right-hand columns per pair = data from the 2020 round of AMIS.

serodiscordant or of unknown serostatus. Proportions are smaller in 2019 than 2020 and in NYC than Atlanta.

**Self-reported impacts of COVID.** Approximately 65% and 42% of respondents in NYC and Atlanta, respectively, reported a decrease in their number of sexual partners because of COVID-19. Impacts on HIV testing were smaller, and not significantly different between cities; 12.1% reported a decrease in Atlanta, and 22.2% in NYC. Very small minorities of men reported that COVID reduced their condom use (6.6% in Atlanta, 8.1% in NYC), with a few saying it increased.

Significant predictors of reporting a direct impact of the pandemic on the number of sexual partners differed between the cities (S4 Table in S1 File). In Atlanta, age 55+ was significantly associated with agreeing with this statement; NYC experienced the same direction of effect, but without reaching significance. Living with one's romantic partner was associated with a much lower probability of reduction in both cities, again only reaching significance in Atlanta. In contrast, NYC was alone in showing a significantly lower level of agreement with the statement among Black MSM relative to White. HIV status, PrEP usage or eligibility (by CDC guidelines), education and income all showed no significant effect in either city.

## Discussion

Numerous works have documented transient declines in sexual behaviors and HIV prevention and care services among US MSM early in the COVID-19 epidemic, and a smaller number have measured subsequent rebounds. Here we expand upon this literature with detailed comparisons of two localities that experienced different COVID-19 waves and policies: New York City and Metropolitan Atlanta. We also consider how changes in each location vary by age.

Our overall results generally comport with three studies considering individual HIV services, especially on the much smaller impact on ART than other services, as expected given that they involve the same data sources [18–20]. However, our comparison of two cities allows

us to consider local differences. We confirmed that indeed, in both locations, ART use showed by far the smallest reductions. This is encouraging, since for men previously diagnosed with HIV, ART needs would not change, and a large, persistent shift here would likely indicate serious access issues. Modeling studies have also shown that reductions in ART access/adherence should have a greater impact on HIV incidence than reductions in other prevention measures [32–34]. The small change we observe appears consistent with a reduction in men initiating ART given reduced testing.

Indeed, testing saw the deepest short-term reduction of any service, dropping to only 13% and 33% of pre-COVID levels in NYC and Atlanta, respectively, before rebounding to ~90% in both. During this trough, positivity increased three-fold in NYC, but not in Atlanta, suggesting that numbers of new HIV diagnoses relative to pre-COVID were actually more similar across cities, although still dramatically reduced. Testing reductions appeared rapidly with COVID-19's outbreak; given that HIV diagnoses often occur months or years after infection [35], it seems likely that a significant portion of that reduction is due to existing infections continuing to go undiagnosed due to rapid shut-down of clinical services rather than simply a decline in new infections from reduced sexual behavior.

PrEP also saw a more substantial decline than ART use; here, the comparison between cities is particularly revealing. When NYC was experiencing a massive first COVID-19 wave, the city faced a dramatic drop in PrEP prescriptions, driven by both decrease in initiations and increase in terminations. Meanwhile, Atlanta was seeing much smaller COVID-19 caseloads, with later and laxer stay-at-home orders. Nonetheless, PrEP use still dropped significantly, albeit with smaller effect size than NYC; this was driven primarily by fewer initiations, and began before stay-at-home orders. By June, PrEP use in both cities ceased dropping, right when COVID-19 cases were *growing* in Atlanta. It thus appears that men in both cities responded primarily to the COVID-19 epidemic's initial appearance overall more than locally-specific conditions. That said, NYC's larger decline presumably reflects both larger local COVID burden and more stringent government mandates, via greater reductions in PrEP indications, greater reduction in availability of in-person clinical services and/or comfort with accessing those services.

Despite rapid expansion of telemedicine early in the pandemic [36,37], none of our HIV service measures had returned to pre-COVID levels by Mar. 2021 except ART prescriptions in Atlanta. While this may reflect lowered perceived service need given reduced transmission opportunities, we observed only small changes in numbers of men reporting no CAI partners in both cities, with similarly small shifts elsewhere in the distribution. Thus, most men continued to report CAI, and roughly 40% in each city reported it with 2+ partners in the last year. Moreover, in both cities we found that the greatest reductions in PrEP and ART prescriptions occurred among young men, while the same age group saw the *least* reduction in partner numbers, including CAI partners. This is concerning, as it may suggest an increase in opportunities for transmission in this age group. We cannot identify individuals across data sources to determine the frequency with which individual men who are discontinuing ART or PrEP continue having many partners. However, further planned analyses with AMIS and other data sources will examine this issue.

The age pattern for self-reporting that COVID-19 caused partner reductions was largely consistent with the direct comparison of partner numbers by age group across wave. The lower reported reduction in partner number for those living with a spouse or romantic partner is understandable, given that for many this may have been their only recent sexual partner to begin with. The large difference in reports by race in NYC, where Black MSM were less likely than others to report a decline, could signal an area of concern; that is, given the sudden drop

in PrEP usage, such a pattern could have created disproportionate short-term opportunities for HIV acquisition among Black MSM, increasing already stark disparities.

Our analysis is consistent with most early-epidemic studies [minus one notable outlier, 14] in finding reductions in partner numbers and increases in the proportion of men reporting one AI partner who is their main partner. A shift to contacting only a main partner (presumably co-resident or in the same COVID-19 "bubble") is clearly beneficial for preventing COVID-19 transmission, but its short-term impact on HIV transmission is more complicated; couples who had been having additional partners might still have the opportunity for transmission of undiagnosed HIV, especially during the acute phase. We note, however, that reports were inconsistent between partner numbers at the population level and self-reported behavior changes; while both show reductions, the former suggest greater change in Atlanta and the latter in NYC. Contributing to this apparent difference may be the fact that AMIS samples were significantly different across years for Atlanta; this would affect the former measure but not the latter, potentially inflating that city's apparent behavior change. Lending further weight to the likelihood that NYC saw more behavior change than Atlanta is one study that found that the prevalence of men avoiding non-co-resident partners varied significantly by state-level COVID-19 diagnoses and length of state stay-at-home-orders [12].

Our testing data reflects a similar pattern to that seen for chlamydia diagnoses nationwide [38], although they looked at positive diagnoses and both sexes, while we measure all tests and only males. They find a somewhat different pattern for gonorrhea and syphilis, with both surpassing pre-COVID diagnosis levels by Mar. 2021. However, this is likely because these two STIs had previously been growing rapidly, while chlamydia had been growing more slowly and HIV diagnoses were declining [39].

Our analysis has many limitations. For clinical data, we could not distinguish MSM from other males; however, MSM comprise most US males living with or at high risk for HIV, and our focus is on relative rather than absolute counts, hopefully minimizing this issue. Our clinical service data sources did not allow us to explore additional predictors like race/ethnicity or socioeconomic status whose role here would also be useful to understand. We also cannot distinguish at the individual level the extent to which reduced service usage reflects reduced need, reduced availability (given clinic closure or staff transfer to COVID-19 activities) or decreased social mobility (by choice or mandate). Some PrEP users may have switched from daily use to on-demand use to conserve medicine while retaining protection; indeed, New York State provided guidance and support for this practice among cis-gender MSM [40]. For time-series analyses, we selected quarters as the unit of analysis *a priori*; more detailed analyses would likely have identified locally-specific change points, as used previously on national data [18,20]. For AMIS, the 2020 survey's retrospective 12-month sexual histories include some time pre-COVID; this should make our year-on-year comparisons to estimate COVID-induced change conservative. Atlanta's larger sample means that similar effect sizes may be significant there and not NYC, a reminder that statistical significance is equivalent to neither effect size nor practical significance. The Atlanta sample differed significantly across years in two key metrics, which could have confounded measures of behavioral changes. Some measures, especially PrEP in Atlanta, were likely changing prior to COVID-19, thus making return to pre-COVID levels an imperfect metric of post-COVID recovery.

These local-level changes during the COVID-19 epidemic undoubtedly have both short- and long-term impacts on HIV incidence and diagnoses. However, estimating that impact will require mathematical modeling that integrates these many effects and simulates resulting potentials for transmission, similar to those during the initial outbreak to predict potential futures [34,41,42]. This work is now in progress, comparing the same two cities using the insights and parameters generated by this analysis. From this we can identify the extent to

which effects of service care disruptions may be cancelled out by short-term behavioral change, both overall and within age group. This will help to clarify the extent to which the Ending the HIV Epidemic goals remain achievable on schedule, or may require re-evaluation in the face of COVID-19, either nationally or in specific locations.

## Supporting information

**S1 File. Supporting information–contains all the supporting text, tables and figures.** (DOCX)

## Acknowledgments

The authors thank the members of the Coalition for Applied Modeling for Prevention (CAMP), especially Monica Trigg, Abby Tighe, Tamika Hoyte, Taiwo Abimbola, Michelle van Handel, Erika Martin, and Eli Rosenberg. Thanks also to Pascale Wortley of the Georgia Department of Public Health and Benjamin Tsoi of the New York City Department of Health and Mental Hygiene, as well as the providers of the data. A special thank you to CAMP's Public Health Advisory Board, especially Jane Kelly, for reviewing this manuscript. Our beloved colleague and co-author, Dawn K. Smith, died before publication of this article; we are forever thankful for her leadership and wisdom.

## Author Contributions

**Conceptualization:** Steven M. Goodreau, Kevin P. Delaney, Dawn K. Smith, Deven T. Hamilton, Karen W. Hoover.

**Data curation:** Steven M. Goodreau, Weiming Zhu, Travis H. Sanchez.

**Formal analysis:** Steven M. Goodreau, Weiming Zhu, Dawn K. Smith, Laura M. Mann.

**Funding acquisition:** Deven T. Hamilton.

**Investigation:** Steven M. Goodreau, Weiming Zhu, Deven T. Hamilton.

**Methodology:** Steven M. Goodreau, Kevin P. Delaney, Weiming Zhu, Laura M. Mann, Karen W. Hoover.

**Software:** Steven M. Goodreau.

**Supervision:** Deven T. Hamilton, Karen W. Hoover.

**Validation:** Steven M. Goodreau.

**Visualization:** Steven M. Goodreau.

**Writing – original draft:** Steven M. Goodreau.

**Writing – review & editing:** Steven M. Goodreau, Kevin P. Delaney, Weiming Zhu, Dawn K. Smith, Laura M. Mann, Travis H. Sanchez, Deven T. Hamilton, Karen W. Hoover.

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
