## [Decision Letter · Decision Letter 0]

6 Dec 2022

PONE-D-22-21830Impacts of COVID-19 on sexual behaviors, HIV prevention and care among men who have sex with men: a comparison of New York City and Metropolitan AtlantaPLOS ONE

Dear Dr. Goodreau,

Thank you for submitting your manuscript to PLOS ONE. After careful consideration, we feel that it has merit but does not fully meet PLOS ONE’s publication criteria as it currently stands. Therefore, we invite you to submit a revised version of the manuscript that addresses the points raised during the review process. Please address the minor comments from the reviewers, with which I agree.

We look forward to receiving your revised manuscript.

Kind regards,

Douglas S. Krakower, MD

Academic Editor

PLOS ONE

Journal Requirements:

3. Thank you for providing documentation stating that ethical approval was not necessary for your study. Please include the information outlined in the documentation in the Methods section of your manuscript.

This work was supported by the U.S. Centers for Disease Control and Prevention’s National Center for HIV/AIDS, Viral Hepatitis, STD, and TB Prevention [cooperative agreement U38-PS004650, PI Erika Martin]. Additional support was provided by a research grant from NIAID [R01 AI138783, PI Samuel Jenness] and a research infrastructure grant from NICHD to the UW Center for Studies in Demography and Ecology [grant number P2C HD042828, PI Sara Curran]. 

As part of the cooperative agreement model, research scientists affiliated with the funder (the US Centers for Disease Control and Prevention) contributed to the design of the study through a series of collaborative meetings. They also approved the decision to submit the manuscript for publication, and CDC staff scientists reviewed and approved the final manuscript. Those individuals who played significant roles in the scientific development of the project are listed as

co-authors.

Reviewers' comments:

Reviewer's Responses to Questions

**Comments to the Author**

1. Is the manuscript technically sound, and do the data support the conclusions?

Reviewer #1: Yes

Reviewer #2: Yes

2. Has the statistical analysis been performed appropriately and rigorously? 

Reviewer #1: Yes

Reviewer #2: Yes

3. Have the authors made all data underlying the findings in their manuscript fully available?

Reviewer #1: Yes

Reviewer #2: No

4. Is the manuscript presented in an intelligible fashion and written in standard English?

Reviewer #1: Yes

Reviewer #2: Yes

5. Review Comments to the Author

Reviewer #1: Overall, this is a wonderfully conducted and well-written paper. I usually have a raft of comments but this rare case, I don't have anything to add. Great job on this, it was a pleasure to read and will be a welcome contribution to the literature.

Reviewer #2: This was an excellent paper modeling the impact of the COVID-19 pandemic and subsequent shelter-in-place ordinances on HIV-related outcomes for men who have sex with men in Atlanta and New York City. The paper was richly and exhaustively detailed, included robust analyses, and provided detailed interpretations of what the findings mean. I commend the authors on an excellent analysis and write-up. I have only a couple minor comments and suggestions:

-I recommend the authors relocate their discussion of the ITS methods from their supplement to the primary paper. Given that interrupted time series analyses are an extension of other longitudinal statistical modeling approaches, it would be helpful for the authors to include this description in the primary paper for the readership.

-In the Results for AMIS, the authors report that substantial fractions of men in Atlanta and NYC self-reported declines in partner numbers as a result of the COVID-19 pandemic. It would be helpful for the authors to present (even unadjusted) characteristics associated with partner reductions, which could help guide inferences regarding whether those who were at greatest risk for HIV reduced partners during the pandemic and would complement their other findings well.

6. PLOS authors have the option to publish the peer review history of their article (what does this mean?). If published, this will include your full peer review and any attached files.

Reviewer #1: No

Reviewer #2: No

---

## [Author Response · Author response to Decision Letter 0]

20 Jan 2023

We thank the editors and the reviewers for the very positive reviews, and minimal comments. Our responses are below:

Reviewer #1: Overall, this is a wonderfully conducted and well-written paper. I usually have a raft of comments but this rare case, I don't have anything to add. Great job on this, it was a pleasure to read and will be a welcome contribution to the literature.

Thank you very much!

Reviewer #2: This was an excellent paper modeling the impact of the COVID-19 pandemic and subsequent shelter-in-place ordinances on HIV-related outcomes for men who have sex with men in Atlanta and New York City. The paper was richly and exhaustively detailed, included robust analyses, and provided detailed interpretations of what the findings mean. I commend the authors on an excellent analysis and write-up. I have only a couple minor comments and suggestions:

Thank you very much!

I recommend the authors relocate their discussion of the ITS methods from their supplement to the primary paper. Given that interrupted time series analyses are an extension of other longitudinal statistical modeling approaches, it would be helpful for the authors to include this description in the primary paper for the readership.

We have moved the explanation of the ITS methods from the supplement to the main paper and integrated it there. We agree this now makes the component of the paper clearer. We believe the reviewer was only referring to the initial description of the methods here, and not also to the presentation of results and their interpretation, given that this would increase the number of figures in the paper to at least 9. If we misunderstood and the reviewer or editor wants more material moved, we are happy to do so. 

In the Results for AMIS, the authors report that substantial fractions of men in Atlanta and NYC self-reported declines in partner numbers as a result of the COVID-19 pandemic. It would be helpful for the authors to present (even unadjusted) characteristics associated with partner reductions, which could help guide inferences regarding whether those who were at greatest risk for HIV reduced partners during the pandemic and would complement their other findings well.

Thank you for suggesting this. This was among the many additional analyses we considered, but struggled with knowing when to stop given the richness of the data sources. We have now added it in to the paper. We opted to put the table itself in the supplement, given the large number of tables and figures already, but put the presentation of results and discussion in the main manuscript. We are happy to move the table to the main manuscript if the editor prefers that.

---

## [Editor Report · Decision Letter 1]

17 Feb 2023

Impacts of COVID-19 on sexual behaviors, HIV prevention and care among men who have sex with men: a comparison of New York City and Metropolitan Atlanta

PONE-D-22-21830R1

Dear Dr. Goodreau,

We’re pleased to inform you that your manuscript has been judged scientifically suitable for publication and will be formally accepted for publication once it meets all outstanding technical requirements.

Kind regards,

Douglas S. Krakower, MD

Academic Editor

PLOS ONE

Additional Editor Comments (optional):

Dear Steven, 

I have reached out to our editorial team to ask about your query regarding a memorial statement for Dawn. As it's been several weeks and I have not received a response, I am absolutely in support of this excellent idea, and you can move ahead with this. If the administrative team that processes your article feels otherwise, then we can let them discuss with you later in the process. Congratulations on your excellent article. Best, Doug
---

## [Editor Report · Acceptance letter]

13 Mar 2023

PONE-D-22-21830R1 

Impacts of COVID-19 on sexual behaviors, HIV prevention and care among men who have sex with men: a comparison of New York City and Metropolitan Atlanta 

Dear Dr. Goodreau:

I'm pleased to inform you that your manuscript has been deemed suitable for publication in PLOS ONE. Congratulations! Your manuscript is now with our production department. 

Kind regards, 

on behalf of

Dr. Douglas S. Krakower 

Academic Editor

PLOS ONE